# Assessment of Biochemical and Neuroactivities of Cultural Filtrate from *Trichoderma harzianum* in Adjusting Electrolytes and Neurotransmitters in Hippocampus of Epileptic Rats

**DOI:** 10.3390/life13091815

**Published:** 2023-08-28

**Authors:** Atef A. Abd El-Rahman, Sally M. A. El-Shafei, Gaber M. G. Shehab, Lamjed Mansour, Abdelaziz S. A. Abuelsaad, Rania A. Gad

**Affiliations:** 1Department of Agricultural Chemistry, Faculty of Agriculture, Minia University, El-Minya 61519, Egypt; sally.ahmed@mu.edu.eg; 2Department of Biochemistry, Faculty of Agriculture, Cairo University, Giza 12613, Egypt; g.shehab@hotmail.com; 3Department of Zoology, College of Science, King Saud University, Riyadh 11451, Saudi Arabia; lmansour@ksu.edu.sa; 4Immunology Division, Zoology Department, Faculty of Science, Beni-Suef University, Beni-Suef 62521, Egypt; elsaad22@science.bsu.edu.eg; 5Department of Pharmacology & Toxicology, Faculty of Pharmacy, NAHDA University (NUB), Beni-Suef 62511, Egypt; raniagad84@yahoo.com

**Keywords:** biochemical, neuroactivities, cultural filtrate, *Trichoderma harzianum*, electrolytes, epileptic rats

## Abstract

Background: Epilepsy is a serious chronic neurological disorder, which is accompanied by recurrent seizures. Repeated seizures cause physical injuries and neuronal dysfunction and may be a risk of cancer and vascular diseases. However, many antiepileptic drugs (AEDs) have side effects of mood alteration or neurocognitive function, a reduction in neuron excitation, and the inhibition of normal activity. Therefore, the present study aimed to evaluate the effect of secondary metabolites of *Trichoderma harzianum* cultural filtrate (ThCF) when adjusting different electrolytes and neurotransmitters in the hippocampus of epileptic rats. Methods: Cytotoxicity of ThCF against LS-174T cancer cells was assessed using a sulforhodamine B (SRB) assay. Quantitative estimation for some neurotransmitters, electrolytes in sera or homogenate of hippocampi tissues, and mRNA gene expression for ion or voltage gates was assessed by quantitative Real-Time PCR. Results: Treatment with ThCF reduces the proliferative percentage of LS-174T cells in a concentration-dependent manner. ThCF administration improves hyponatremia, hyperkalemia, and hypocalcemia in the sera of the epilepticus model. ThCF rebalances the elevated levels of many neurotransmitters and reduces the release of GABA and acetylcholine-esterase. Also, treatments with ThCF ameliorate the downregulation of mRNA gene expression for some gate receptors in hippocampal homogenate tissues and recorded a highly significant elevation in the expression of SCN1A, CACNA1S, and NMDA. Conclusion: Secondary metabolites of Trichoderma (ThCF) have cytotoxic activity against LS-174T (colorectal cancer cell line) and anxiolytic-like activity through a GABAergic mechanism of action and an increase in GABA as inhibitory amino acid in the selected brain regions and reduced levels of NMDA and DOPA. The present data suggested that ThCF may inhibit intracellular calcium accumulation by triggering the NAADP-mediated Ca^2+^ signaling pathway. Therefore, the present results suggested further studies on the molecular pathway for each metabolite of ThCF, e.g., 6-pentyl-α-pyrone (6-PP), harzianic acid (HA), and hydrophobin, as an alternative drug to mitigate the side effects of AEDs.

## 1. Introduction

People of all ages are susceptible to the chronic, noncommunicable brain disorder known as epilepsy. As one of the most prevalent neurological conditions worldwide, epilepsy affects over 50 million people worldwide, with nearly 80% of them living in poor countries. Seizure’s aftereffects, such as premature mortality or residual disability, might result in health loss [1]. Although the epilepsy burden decreased from 1990 to 2016, it is considered a vital reason for disability and mortality. The more responsible reasons for epilepsy are still unknown in approximately 50% of global patients, even though many known mechanisms are well-defined to underline the disease [2].

Drugs can be used to treat epilepsy; for instance, phenobarbital has a good response rate in 75% of patients [3]. Receiving sodium valproate medication for a year may cure approximately 42% of patients from seizures and decrease the seizure frequency in approximately 84% of patients [4]. However, many antiepileptic drugs (AEDs) have negative consequences on brain activity, such as changes in mood or neurocognitive performance [3,5], a reduction in neuron excitation, and the inhibition of normal activity. Therefore, many researchers are focused on finding more effective or alternative drugs to mitigate the different side effects of AEDs [6].

One of the most natural products released from the fungi Ganoderma lucidum has been used to treat various human diseases in East Asia, e.g., in cases of hypercholesterolemia, hypertension, hepatitis, and cancer [7,8]. The fungal spores of Ganoderma lucidum have anti-epileptic activities in both in vivo and in vitro experiments [9,10,11].

Gamma-aminobutyric acid (GABA) is the primary inhibitory neurotransmitter in the central nervous system (CNS) [12]. In status epilepticus, the harmony between excitement and inhibition is frequently upset. It is possible for GABAergic inhibitory systems to deteriorate, which would result in excessive excitatory activity and an inability to stop the seizure. A reduced GABA concentration or compromised GABAergic transmission may be linked to this imbalance. During status epilepticus, the balance between excitation and inhibition is frequently disrupted and GABAergic inhibitory systems may deteriorate, leading to excessive excitatory activity and an inability to stop the seizure [13,14]. Seizure-induced plasticity of synaptic GABAA receptors present in principal neurons of the hippocampus is also a factor in the development of pharmaco-resistance in status epilepticus [14]. Moreover, Acetylcholine-esterase (AChE) is an enzyme that breaks down the neurotransmitter acetylcholine (ACh) into choline and acetate. AChE upregulation has been observed in the temporal lobe structures of rats that have experienced status epilepticus (SE) [15].

Low blood sodium levels or severe hyponatremia can reduce cerebral blood flow and eventually cause centrally mediated respiratory depression, which can result in seizures. Hyponatremia can result in cerebral edema and neuronal swelling, which can increase intracranial pressure and lead to seizure activity [16,17]. High blood potassium levels or hyperkalemia infrequently result in symptoms in the central nervous system (CNS), and seizures are not frequently associated with this condition. Severe hyperkalemia, however, can have cardiac side effects such as arrhythmias that could cause seizures afterward [18]. Low blood calcium levels or hypocalcemia can enhance neuronal excitability and increase the risk of seizures in susceptible people. The balance between excitatory and inhibitory neurotransmission is critical for stabilizing neuronal membranes, and if it is out of balance, seizures may result [18,19].

However, there is no research on *Trichoderma*’s impact on the expression of several neurotransmitters and electrolytes in neurons with epilepsy. Generally, natural products like fungi belonging to the *Trichoderma* genus are well-known producers of secondary metabolites [20]. Moreover, different species of fungi release many secondary metabolites (SMs), with relatively small molecules, typically less than 3 kDa, and are chemically different active natural compounds. Many fungal SMs have potential medicinal applications, e.g., penicillin and cephalosporins antibiotics, anti-hypercholesterolemic agents lovastatin and compactin, and immunosuppressant cyclosporine [21]. The most important biological activities of peptaibols of *Trichoderma* were previously reported [22].

Secondary metabolites derived from various *Trichoderma* fungal isolates consist of peptaibiotics, siderophores, and diketopiperazines, including gliotoxin and gliovirin, polyketides, terpenes, pyrones, and isocyanate metabolites as examples of non-ribosomal peptides. Some of Trichoderm’s metabolites have significant antifungal activity (e.g., pyrone, koninginins, and nitrogen heterocyclic compounds). Pyrones are commonly purified pyrone 6-pentyl-2H-pyran- 2-one (6-pentyl-pyrone) from *Trichoderma harzianum*, *T. viride*, *T. atroviride, and T. koningii* [23]. Moreover, *T. harzianum* releases complex pyranes named Koninginins [24,25]. Nitrogen Heterocyclic Compound (Harzianopyridone) metabolites were isolated from *T. harzianum* and are considered a potent active antibiotic against *Pythium ultimum, G. graminis* var. *tritici tritici* [26], *R. solani,* and *Botrytis cinerea* [27]. Additionally, *Pythium irregulare*, *Sclerotinia sclerotiorum*, and *R. solani* are all susceptible to the antibiotic properties of harzianic acid, which is isolated from *T. harzianum* [28]. Other SMs called azaphilones are natural products that are isolated from the liquid culture of *T. harzianum* T22 [26]. Butenolides and Hydroxy-Lactones, e.g., harzianolide, are isolated from different strains of *T. harzianum* and have antifungal activities against several phytopathogenic agents [26]. Other Peptaibols are linear peptides rich in non-proteinogenic amino acids (i.e., α-aminoisobutyric acid and isovaline). They are acetylated at the N-terminal group and the C-terminus as an amino alcohol (i.e., phenylalaninol, valinol, leucinol, isoleucinol, or tryptophanol) [29].

However, peptaibols are the more studied class for physicochemical aspects and biological activities [22]. In addition, peptaibols have many hemolytic properties, e.g., alamethicin effects on mast cells of rats, lymphocytes of bovine and mice, and erythrocyte lysis [30]. Peptaibols released from *T. harzianum* inhibit the motility of sperm cells of domestic pigs in vitro and correlate with human cell toxicity [31].

Moreover, approximately seven new peptaibols (trichokindins I–VII) were found in *T. harzianum*. They were capable of the induction of Ca^2+^-dependent catecholamine secretion from bovine adrenal medullary cells [32]. Further, they characterized the anti-tumor activities of the new compound on CM126 and HT-29 cell lines, affecting the HT-29 cell cycle at the S phase.

Non-volatile (such as peptaibols) and volatile (such as simple aromatic metabolites, terpenes, the isocyanic metabolites, certain polyketides, butenolides, and pyrones) SMs released by *T. harzianum* are divided into several classes based on their chemical structure. The *T. harzianum* sample included approximately 278 identified fungal metabolites. The volatile substances identified in the culture samples belong to the following compound classes: Monoterpenes, sesquiterpenes, monoterpene alcohols, pyrones (lactones), furanes, alkanes, and alcohols. The main components from *T. harzianum* identified were hydrocarbons, fatty acids, alcohol, and derivatives of benzene, such as cyclohexane, cyclopentane, and other substances that were discovered among the volatile metabolites. 6-Pentyl-alpha-pyrone was one of the most prevalent metabolites that was found (6-PP). This is the most significant volatile chemical produced from pyrone. Before being identified as a natural product, this substance, a harmless flavoring agent, was chemically manufactured for industrial use. Numerous biological effects of 6-PP have been observed, including a reduction in the mycotoxin deoxynivalenol generation by *Fusarium graminearum* and the exertion of antifungal characteristics by slowing the growth of *Rhizoctonia solani* and *F. oxysporum* mycelium [33].

The present study aimed to investigate the potential mechanism of the cultural filtrate of *T. harzianum* (ThCF) in the treatment of epilepsy or the mitigation of the side effects of antiepileptic drugs. This aim was achieved via the estimation of different electrolytes and neurotransmitters and their mRNA gene expression for their gates in the hippocampal neurons of the epilepticus model treated with SMs of fungal *T. harzianum*.

## 2. Materials and Methods

### 2.1. Animals

Thirty-two white mature male Sprague Dawley rats (weighing between 150 and 180 g) were procured from the National Research Institute’s animal house in Eldokki, Giza, Egypt. They were kept in private metabolic cages in a 23 °F, 55 °F humidity environment with a 12-h light/dark cycle. Water and food were freely available.

### 2.2. Drug and Chemicals

Acros Organics provided 99% pilocarpine hydrochloride (PILO) (Fair Lawn, NJ, USA). Valproic acid was also used (Depakine Chrono^®^500 mg, Sanofi Aventis CO, Paris, France).

### 2.3. Preparation of the Fungal Cultural Filtrates of Trichoderma Isolates

*Trichoderma harzianum* was generously donated by the Plant Pathology Department, Faculty of Agriculture, Minia University, El-Minia, Egypt, for this study. Briefly, *T. harzianum* was cultivated on PDA (potato dextrose agar) for 10 days at 28 °C in Petri plates until the observation of conidia production. After *T. harzianum* sporulation in plates containing PDA, the fungal isolate was cultivated on potato-glucose liquid media for 168 h on an incubator shaker (28 °C, 140 rpm). The culture medium was made with 200 mL of potato decoction, 20.0 g of glucose, and 800 mL of distilled water. After 168 h of shaking, the culture broth containing *Trichoderma* spores was filtered using Millipore filter paper (0.22 μm) to obtain the *Trichoderma* cultural filtrate devoid of spores. Under vacuum, the supernatant was recovered and dried. The obtained crude extract was weighed, kept at 20 °C until use, and then re-suspended in sterile phosphate-buffered saline (PBS, pH 7.4). The dose was estimated according to other treatments [34], which had an LD50 of 1087 mg/kg in females and 644 mg/kg in male rats.

### 2.4. Cell Culture and Cytotoxicity Assays

The colorectal cancer cell line (LS-174T) was kindly provided by Nawah Scientific Inc. (Mokatam, Cairo, Egypt). The RPMI medium supplemented with 100 mg/mL streptomycin, 100 units/mL penicillin, and 10% heat-inactivated fetal bovine serum was used to sustain cells at 37 °C in a humidified 5% (*v*/*v*) CO_2_ atmosphere.

A cell cytotoxicity assay using Sulforhodamine B (SRB) was employed to detect viability or study the drug cytotoxicity. This approach is based on the ability of SRB, a bright pink amino xanthene dye, to stoichiometrically bind to proteins in mildly acidic conditions before being extracted under basic conditions. As a result, the amount of bound dye can be used as a stand-in for cell mass, which can then be extrapolated to measure cell proliferation at 565 nm [35,36].

Briefly, cell viability was assessed by an SRB assay. Aliquots of a 100 μL cell suspension (5 × 10^3^ cells) were placed in 96-well plates and incubated in complete media for 24 h. Cells were treated with another aliquot of 100 μL media containing ThCF at various concentrations ranging from (0.01, 0.1, 1, 10, and 100 μg/mL). After 72 h of drug exposure, cells were fixed by replacing media with 150 μL of 10% TCA and incubated at 4 °C for 1 h. The TCA solution was removed, and the cells were washed 5 times with distilled water. Aliquots of 70 μL of the SRB solution (0.4% *w*/*v*) were added and incubated in a dark place at room temperature for 10 min. Plates were washed 3 times with 1% acetic acid and allowed to air-dry overnight. Then, 150 μL of TRIS (10 mM) was added to dissolve the protein-bound SRB stain; the absorbance was measured at 540 nm using a BMG LABTECH^®^- FLUOstar Omega microplate reader (Ortenberg, Germany).

After the cell culture and cell growth studies, the following formula is used in toxicity testing to assess the impact of substances on cell viability and proliferation according to Orellana, E. A. and A. L. Kasinski [37] and Patel, S., N. Gheewala, A. Suthar, and A. Shah [38] as follows: % Cell growth = (At − Ab)/(Ac − Ab) × 100; where At = Absorbance value of treated cells; Ab = Absorbance value of blank cells; Ac = Absorbance value of control cells; % Cell growth inhibition = 100 − % Cell growth.

### 2.5. Induction of Epilepsy

Experimentally induction of epilepsy was performed [39]. Briefly, the experimental rats received intraperitoneal injections of methylscopolamine (1 mg/kg) for 30 min before treatment with 300 mg/kg of pilocarpine hydrochloride. After each injection, convulsive behavior was observed for 30 min, and resultant seizures were scored according to [40] as follows: Stage 0, no response; stage 1, ear and facial twitching; stage 2, convulsive waves axially through the body; stage 3, myoclonic jerks and rearing; stage 4, clonic convulsions with the animal falling on its side; stage 5, repeated severe tonic–clonic convulsions or lethal convulsions. The animals were considered to be kindled after receiving pilocarpine hydrochloride injections and having exhibited at least three consecutive stage 4 or 5 seizures [41]. Rats experiencing lethal convulsions and those who did not develop kindling were excluded from the study. Diazepam (4 mg/kg, i.p.) was given as needed, every 20 min, to stop seizures. The identical procedure was used to treat the control rats, except that phosphate-buffered saline (PBS, pH 7.4; 0.2 mL/rat) was administered in place of pilocarpine, and diazepam was administered one hour later. The experimental rats were grouped (8 rats/group) as follows:(1)Control group (C): Eight rats were simply given the regular diet, unrestricted access to sterile water, and 0.2 mL/rat of PBS (pH 7.4) orally at intervals parallel to those of the treated groups.(2)Positive epileptic group (EP): Treated with 300 mg/kg of pilocarpine hydrochloride injection as described previously.(3)Valproic acid (Depakine^®^) treated group (EP-VPA): Received a standard market drug (EP-VPA); eight rats/group).(4)ThCF treated group (EP-ThCF): Injected with 300 mg/kg PILO (as described previously), then fed orally with *Trichoderma* cultured filtrate (ThCF) using intragastric intubation at a dose of 400 mg/kg b.wt/day for continued four weeks (i.e., less than LD50 [34]).

### 2.6. Blood and Tissue Sampling and Neurotransmitters and Electrolytes

At the end of the experiment, rats were euthanized and sacrificed. Immediately, blood was collected into a sterile vacutainer tube. Sera were collected and prepared by centrifugation at 3500× *g* for 15 min, transferred into sterilized tubes, and stored at −20 °C until being used for electrolytes and neurotransmitters estimation, and further mRNA expressions of different ion and voltage gates.

Slices of the hippocampus were removed from the brain and placed on ice-filled plates to measure the electrolytes and neurotransmitters. In a nutshell, portions of hippocampal tissue were homogenized at 1000× *g* for eight up-and-down strokes in chilled PBS (pH 7.4) using a Potter–Elvehjem homogenizer (Braun, Melsungen, Germany). Following filtration, the homogenate was centrifuged in a Beckman TJ-6 centrifuge at 600× *g* for 10 min at 4 °C (Beckman Instruments; Munich, Germany). The clear supernatant was preserved at −80 °C until several investigations were evaluated. The total protein in the supernatant was calculated as gm/tissue [42].

Neurotransmitters were performed according to the purchased manufacturer’s guide. MyBioSource Co. (San Diego, CA, USA) assay kits were used for the quantitative detection of L-dihydroxyphenylalanine (L-DOPA; µg/mL), epinephrine (ng/mL), Norepinephrine (pg/mL); glutamate (µg/mL), and gamma-aminobutyric acid (GABA, pg/mL) using kits with catalog no. MBS9357024, MBS031232, MBS269993, MBS047402, and MBS269152, respectively. Acetylcholinesterase (U/L) activity was performed according to the instruction manual of QuantiChrom^TM^ Assay of BioAssay Systems Kit (Cat. No: DACE-100, San Diego, CA, USA). According to the instruction manual of MyBioSource, the concentration of electrolytes (mMol/L) in either the serum or homogenate of hippocampi, e.g., sodium (Na^+^, MBS846620) or potassium (K^+^, MBS8243253), was determined. In addition, calcium (Ca^2+^) and chloride (Cl^−^) levels were determined according to the quantitative colorimetric instruction manual of QuantiChrom^TM^ assays of BioAssay Systems as DICA-500 and DICL-250, respectively.

Regulations of different mRNA gene expression for SCN1A (Na^+^ channel), KCNJ2 (K^+^ channel), CACNA1S (Cav1.1) (Ca^2+^ channel), CLCN2 (GABA-receptor), and N-methyl-D-aspartate (NMDA; Glutamate Receptor) were estimated by quantitative Real-Time PCR (qPCR). Total RNA was separated from brain tissue samples according to the method of Chomczynski and Sacchi [43] using a Qiagen tissue extraction kit (USA) to generate cDNA and perform RT-PCR. The quality and quantity of the resultant PCR products were analyzed using a UV spectrophotometer. Ratios between 1.8 and 2.0 for A260/A280 were accepted as indicating pure RNA, so were used in the next steps. Next, we converted the purified RNA to cDNA and amplified the cDNA using a My Taq One-Step RT-PCR Kit (Bioline, Meridian Bioscience, Memphis, TN, USA) in the presence of specific primers (LGC Biosearch Technologies, Petaluma, CA, USA) (Table 1). All experiments were performed in triplicate independently.

### 2.7. Statistical Analysis

The data were examined using the Tukey–Kramer post-hoc analysis approach to compare different groups with one another. The mean and standard deviation were used to express the results (SD). For all data, the statistical significance interval is defined as *p* < 0.05. Statistical Package for Social Science (SPSS) version 20 was used to examine each result (IBM Corp., Armonk, NY, USA, 2011).

### 2.8. Ethics Committee Approval

The Animal Research Ethics Committee at Beni-Suef University granted consent for all procedures, which were carried out in accordance with the rules for the care and use of laboratory animals and assigned an approval number: BSU/FS/2021/021/138. All procedures carried out in the current study were approved by the ethical committee for the care and use of animals, microorganisms, and living cell cultures in education and scientific research, Faculty of Agriculture, Minia university, El-Minya, Egypt and assigned the following approval number: MU/FA/016/12/22.

## 3. Results

The present results obtained by the Sulforhodamine B (SRB) cell cytotoxicity assay against colorectal cancer (LS174T) cell lines are summarized in Figure 1. The results exhibited a concentration-dependent decrease in the cell viability of both LS174T following 72 h exposure to different concentrations of ThCF (0, 0.01, 0.1, 1, 10, and 100 μg/mL).

Moreover, assessing the cytotoxic effects of culture filtrate *Trichoderma harzianum* (ThCF) against the colorectal cancer cell line (LS-174T) is calculated in Table 2. The results revealed that the percentage of LS-174T cell growth was assessed after being exposed to various concentrations of ThCF for 72 h at concentrations of 0, 0.01, 0.1, 1, 10, and 100 µg/mL, and the results are displayed in Table 1. The proportion of LS-174T cell proliferation percentage was statistically significantly reduced (*p* < 0.05) by ThCF administration in a concentration-dependent manner. As shown in Table 1, ThCF was found to be cytotoxic for LS-174T cells when it was administered for 72 h at concentrations of 0.01 µg/mL and higher doses up to 100 µg/mL. The LS-174T cell proliferation percentages were 100.00 ± 0.00, 89.91 ± 1.03, 86.22 ± 1.26, 82.43 ± 1.55, 80.35 ± 0.43, and 75.81 ± 0.71% at concentrations of 0, 0.01, 0.1, 1, 10, and 100 µg/mL of ThCF, respectively. The present data recorded that IC50 was >100 μg/mL.

In addition, after 72 h of incubation with ThCF, the percentage of cell growth inhibition in LS-174T cancer cells was calculated. The results are displayed in Table 2. The findings indicated that the LS-174T cells were sensitive to ThCF, and they recorded percentages of cell proliferation inhibition of 00.00 ± 0.00, 10.09 ± 1.03, 13.78 ± 1.26, 17.57 ± 1.55, 19.64 ± 0.43, and 24.19 ± 0.71% at concentrations of 0, 0.01, 0.1, 1, 10, and 100 μg/mL of ThCF, respectively, compared to the control cells. ThCF inhibited cell growth in LS-174T cells by a moderate and high proportion at dosages of 1 and 100 μg/mL, respectively. At doses of 1 and 10 µg/mL of ThCF, no significant (*p* < 0.05) differences in cell growth or cell inhibition percentages were found.

Regarding changes in electrolytes either in sera or tissue homogenate, sodium ions (Na^+^) were significantly increased as hyponatremia (*p* > 0.05) in the serum of EP individuals (130.65 ± 1.37 mM/mL) compared to the healthy group (139.29 ± 0.56 mM/mL). However, significant hypernatremia (*p* > 0.05) was encountered in the hippocampal homogenate tissue of the epileptic group (29.47 ± 0.57 mM/mL) as compared with controls (22.62 ± 0.62 mM/mL Tissue). On the other hand, both Valproic acid (EP-VPA) and *T. harzianum* treatments (EP-ThCF) returned Na^+^ levels in sera (137.10 ± 1.70 and 137.36 ± 1.19, respectively) to near control levels. Moreover, a significant improvement (*p* > 0.05) in the Na^+^ level was noticed in the hippocampal homogenate tissue after EP-VPA or EP-ThCF treatments (21.89 ± 1.83 and 22.86 ± 2.59 mM/mL tissue, respectively) in comparison to EP (Table 3).

Currently, the data recorded significant (*p* > 0.05) hyperkalemia (high level of K^+^) in the sera of EP individuals (6.21 ± 0.26 mM/mL) in comparison to all groups: Control, EP-VPA, and EP-ThCF, which have serum K^+^ levels of 4.63 ± 0.31; 6.01 ± 0.59, and 4.65 ± 0.65 mM/mL, respectively (Table 2). On the other hand, K^+^ levels in the homogenate tissues of the hippocampus of EP recorded significant hypokalemia (*p* > 0.05) (66.52 ± 5.85 mM/mL Tissue), as compared with the control group (94.69 ± 7.50 mM/mL Tissue) and EP_DK group (78.90 ± 3.29 mM/mL Tissue), while EP-ThCF treatment ameliorates the K^+^ level in homogenate tissues with 90.18 ± 5.46 (Table 2).

Highly significant hypocalcemia (*p* > 0.001) in sera Ca^2+^ levels (0.37 ± 0.07 mM/mL) of the EP group (Table 2) was observed compared to controls (1.05 ± 0.18 mM/mL), while treatments with EP-VPA and EP-ThCF recorded significant re-elevation (*p* > 0.05) in serum Ca^2+^ (0.81 ± 0.05 and 0.81 ± 0.14 mM/mL, respectively). Significant one-fold hypercalcemia (*p* < 0.001) was noticed in the hippocampal homogenate tissue of EP (0.61 ± 0.02 mM/mL Tissue) in comparison to controls (0.30 ± 0.05 mM/mL Tissue), while EP-VPA and EP-ThCF individuals showed a noticeable amelioration of the level of Ca^2+^ (0.29 ± 0.02 and 0.34 ± 0.02 mM/mL Tissue, respectively).

Significant hyperchloremia (*p* < 0.01) was noticed in the EP group in sera (110.05 ± 4.43 mM/mL) and hypochloremia (*p* < 0.01) in the hippocampal homogenate tissues (79.86 ± 6.97 mM/mL Tissue) of the EP group, comparable with controls (103.49 ± 2.17 mM/mL in serum and 123.72 ± 3.37 mM/mL in tissue) (Table 2). In addition, data revealed that Cl- was significantly recovered (*p* > 0.001) either in the serum of EP-VPA (103.73 ± 1.40 mM/mL) or in its hippocampal homogenate tissue (99.99 ± 13.16 mM/mL Tissue). Also, Cl- levels were ameliorated after treatment of EP-ThCF and showed significance (*p* < 0.001) in serum (104.11 ± 2.57 and 105.60 ± 8.25 mM/mL, respectively).

Gene expression for SCN1A mRNA (Na^+^) showed a highly significant increase (*p* < 0.001) in EP’s hippocampal tissues (1.52 ± 0.15 relative to the control) (Figure 2), while both EP-VPA and EP-ThCF groups showed significantly decreased levels (F = 180.595 and *p* < 0.0001) of the mRNA gene expression of SCN1A (Na^+^), as they recorded a relatively low expression 0.76 ± 0.05 and 0.79 ± 0.05, respectively, compared to the EP or control groups (Table 2).

The gene expression of the potassium gate channel KCNJ2 (K^+^) recorded a highly significant decrease (0.36 ± 0.04 relative to the control, with F value= 163.387 and *p* < 0.001 in EPa), unlike in the EP-ThCF group, which showed a significant increase (*p* < 0.001) and recorded 0.61 ± 0.05 comparable to EP and showed non-significant variations with the EP-VPA treated group (0.61 ± 0.05) (Figure 2).

Significantly, CACNA1S Ca^2+^ (Cav1.1) was increased (*p* < 0.001) in EP rats and recorded 7.22 ± 0.37 relative to the control, while treatment with ThCF and EP-VPA ameliorated such expression and recorded 2.87 ± 0.55 and 3.33 ± 0.51, respectively, still significantly more than controls (with F value = 330.582 and *p* < 0.001) (Figure 2).

Regarding the ligand-gated ion channel and GABA receptor, the present data showed that their gene expression of mRNA of CLCNC recorded significant downregulation (with F value = 229.405 and *p* < 0.001) in EP animals (0.15 ± 0.03) comparable to controls (1.03 ± 0.02). Moreover, mRNA gene expression of CLCNC in EP-VPA and EP-ThCF groups showed a significant elevation (*p* < 0.001) and recorded 0.59 ± 0.07 and 0.71 ± 0.11, respectively, as compared with the EP relative to control expression (1.03 ± 0.02) (Figure 2). On the other hand, the present data recorded that mRNA gene expression of NMDA (as a glutamate receptor) was highly upregulated in EP (7.33 ± 1.11 than controls 1.02 ± 0.01 (with F value = 185.172 and *p* < 0.001). However, this expression was significantly decreased (*p* < 0.05) after treatment with EP-VPA ThCF, and recorded 3.00 ± 0.14 and 2.72 ± 0.46, respectively (Figure 2).

Concerning changes in the neurotransmitters in hippocampal homogenate tissues, e.g., DOPA (Ug/protein tissue), epinephrine (pg/mg protein tissue), norepinephrine (pg/mg protein tissue), and glutamate (ng/mg protein tissue), Figure 2 illustrate a very highly significant increase, recording 36.40 ± 3.39 (F = 139.238 and *p* < 0.001); 137.34 ± 7.09 (F = 88.048 and *p* < 0.001); 213.41 ± 14.84 (F = 87.346 and *p* < 0.001), and 16.11 ± 2.08 (F = 117.106 and *p* < 0.001)), respectively. Treatments with EP-VPA or ThCF improved this significant change and returned them to near control levels as illustrated in Figure 3. On the other hand, both GABA (Figure 3D) and acetylcholine-esterase (Figure 3F) were significantly improved to the control level (56.28 ± 4.18 and 94.96 ± 6.48, respectively) either by treatment with EP-VPA (43.79 ± 3.61 with F = 138.638 and *p* < 0.001) and 50.22 ± 7.72 with F = 89.662 and *p* < 0.001), respectively) or ThCF (72.81 ± 6.53 and 78.40 ± 9.91, respectively) (Figure 3).

## 4. Discussion

Previously, *Trichoderma* spp. has cytotoxic activities against human cervical and breast cancer cell lines and produced many anticancer agents [44] as many of the extracellular enzymes are secreted by *Trichoderma* and regulate its biocontrol activities [45,46,47]. Some of such extracellular enzymes have anticanrinogenic activity [48]. L-lysine oxidase as one of *T. viride* metabolites has antineoplastic activity [49]. ThCF has cytotoxic activities and altered the cellular morphology of human cervical and breast cancer cell lines (HeLa and MCF-7) in a concentration-dependent manner [44]. L-methioninase, one of *T. harzianum’s* secondary metabolites, inhibited the proliferation of MCF-7 and HEPG2 cell lines [50]. Cell cycle arrest, cell cycle delay, and the apoptotic process are three mechanisms to inhibit the proliferation of cancer cells. The findings of a recent study suggest that *Trichoderma reesei* ethyl acetate extract has the potential to operate as an anti-cancer gene because of its ability to promote apoptosis [51]. Moreover, the acute intraperitoneal toxicity/pathogenicity of *Trichoderma* spp. was investigated previously by Rees Rees, P. [34], who intraperitoneally administered an injection of Trichodex^®^, which contains *Trichoderma harzianum* Rifai strain T-39. He revealed that in male rats, the LD50 for the *Trichoderma harzianum* was found to be 644 mg/kg, while it was 1087 mg/kg in female rats. The combined male and female rat results were 806 mg/kg. In addition, there were no signs of substantial deleterious effects at the lowest dose of 1.5 × 10^7^ cfu/animal. This trial was considered acceptable as an alternative to an intravenous study using fungi as active pesticides.

Briefly, treatment with the secondary metabolites of (ThCF) improves hyponatremia, hyperkalemia, and hypocalcemia in the sera of an epilepticus model, rebalances the elevated levels of many neurotransmitters, e.g., DOPA, epinephrine, norepinephrine, and glutamate, and reduces the release of GABA and acetylcholine-esterase. Also, treatments with EP-VPA or ThCF ameliorated the downregulation of mRNA gene expression for some gate receptors in hippocampal homogenate tissues, e.g., KCNJ2 and CLCNC, which were elevated parallel with the gene expression of SCN1A, CACNA1S, and NMDA’ gates.

The recorded high extracellular K^+^ accumulation in epilepticus hippocampus tissues occurred in parallel with others [52,53]. Glial K^+^ loss of its conductance was explained by the high-level accumulation of extracellular K^+^ and correlated to epileptic neuronal hyperexcitability [53]. Seizures and mood disorders are associated with mutations in KCNJ2 that encode the inwardly rectifying K^+^ channel (Kir2.1) [54]. Highly expressed Kir2.1 channels in all brain areas are implicated in cognition and mood disorders, e.g., the hippocampus [55]. More than 70 genes encoded membrane proteins such as K^+^ channels to fine-adjust the neuronal and non-neuronal electrical activity [52]. Intrinsically, K^+^ channels regulate neuronal excitability, synaptic plasticity, and cell differentiation [56]. Therefore, any dysfunction or decoding of such channels caused critical functional impairment of the neurophysiological processes of neural networks [57]. The low expression of the pore-forming K^+^ channel may cause neonatal epileptic encephalopathy and impaired neurodevelopment [58]. They mentioned a faster action potential repolarization after hyperpolarization and enhancement of Ca^2+^-activated K^+^ channels’ functions. The recorded imbalance in extracellular K^+^ affects the regulation of cardiovascular and neuromuscular activities and causes arrhythmias or paralysis [59].

Calcium (Ca^2+^) is a second-messenger molecule and plays different roles in many of the neuronal cascades, e.g., muscle contraction, neuro potentiation, cell proliferation, and programmed cell death [60]. Atypical absence or kinetic seizures are characterized by acute hypocalcemia, which alters the tonic–clonic focal motor and alters mental activity [61]. Therefore, the homeostasis of Ca^2+^ content is important for the prevention and treatment of epilepsy [11]. The recorded high Ca^2+^ content in the present hippocampus tissues was parallel to the results of others [11], who reported an increase in the fluorescence intensity of Ca^2+^ in neurons of an epileptic hippocampus compared to normal neurons. Moreover, results showed that fungal polysaccharides of G. lucidum can reduce the epileptic hippocampus Ca^2+^ by suppressing the overload of calcium to prevent episodes of epilepsy. The present data recorded an overload of Ca^2+^ and more influx into hippocampus tissues, leading to the onset of epilepsy [11].

The prevention of epilepsy can be achieved by using chelating agents to reduce intracellular calcium (Ca^2+^i) or decrease extracellular Ca^2+^ environments [62]. Moreover, tissue hypercalcemia may be due to Ca^2+^ influx, which activated SK/KCa2 channels in excitable cells through both nicotinic acetylcholine and ionotropic receptors (e.g., NMDARs) [63]. Opened SK/KCa2 channels cause more hyperpolarization and imbalance of Ca^2+^ in the hippocampus homogenate tissue [64]. A conformational change in SK/KCa2 channels will cause more K^+^ influx through the channel pore [65].

Furthermore, activation of the NMDA receptor induced different alterations in the neuronal excitability in hippocampal neuronal culture [66] and hippocampal slice [67] as models of epilepsy. Therefore, the induction of epilepsy is positively correlated with elevated levels of intracellular Ca21 (Ca21]i) [62]. The fungal spores of G. lucidum have anti-epileptic activities both in vivo and in vitro [10,11]. Recently, it was reported that the secondary metabolites of fungal Ganoderma lucidum have been shown to increase GABA, decrease glutamate [68], inhibit the expression of both NF-kB and N-Cadherin, and stimulate Cav-1 and neurotrophin-4 expression [68].

Seven new peptaibols (trichokindins I–VII) were found in *T. harzianum*. They are capable of potentiating bovine adrenal medullary cells to secrete Ca^2+^-dependent catecholamine [32]. Therefore, the recorded inhibition of intracellular calcium accumulation may be affected by the peptaibols’ contents of *T. harzianum*. Further investigations are required to explain the mechanism of such activity, therefore, the present study suggested the possible role of chelating reactive oxygen species (ROS), in improving and protecting membrane stability [7,8].

Research pointed out that the hippocampus has many expressed synaptic and extra-synaptic GABAA receptors [69]. These receptors are linked to negative emotions like depression, anxiety, catastrophizing, and stress [56]. The present data showed that treatment with ThCF ameliorated levels of GABA amino acids in the hippocampus area. Significantly, pyrone 6-PP (pyrone-6-pentyl-2H-pyran-2-one) released as secondary metabolites of *T. harzianum* can initiate more production of acetylcholine and GABA to protect plants from pathogens [70]. Therefore, 6-PP is regarded as a GABA-A receptor activator [71], as it has a similar effect on BZDs as anxiolytic drugs, prompting increased chloride conductance after elevation of the affinity of the GABA-receptor [72], resulting in the suppression of the neurotransmission and sedative and anxiolytic effects [73].

Moreover, high levels of GABA in the amygdala, hippocampus, and prefrontal cortex regions may associate with an anxiolytic action [74]. Many clinical investigations revealed a positive correlation between GABA and anxiety symptoms [73]. Finally, regarding the GABAergic mechanism of action and the rise in GABA as an inhibitory amino acid in the targeted brain region, the current data explained the role of ThCF in anxiolytic-like activity.

Effective anticonvulsant activity may target more than one mechanism of molecular pathways, e.g., interactions with components of the synaptic release machinery, enhancement of GABAergic transmission, blockade of ionotropic glutamate (NMDA and AMPA) receptors, or modulation of voltage-gated ion channels and the SV2a protein [75]. It is worth mentioning that regardless of the mechanism of action, they all act to reduce hyperexcitability by either decreasing excitatory or enhancing inhibitory neurotransmission [76]. The establishment and control of the excitability of CNS nerves appear to depend heavily on voltage-dependent sodium and calcium channels. They are the most prevalent targets for anticonvulsant medications that are currently on the market, such as phenytoin, carbamazepine, lamotrigine, oxcarbazepine, and lacosamide [64]. On the other hand, several mechanisms of action, including glutamate (NMDA) inhibition and GABA potentiation, are linked to valproic acid, in addition to the blockade of both Na^+^ and T-type calcium channels [75].

Epilepsy is induced by the activation of NMDA receptors [77], a family of L-glutamate receptors, which leads to high levels of extracellular glutamate and the generation of ROS in neurons [78]. The present data were in parallel with others [62], who proved that the long-term (2–3 h) activation of the NMDA receptor-Ca21 transduction pathway can induce in vitro emergence of spontaneous recurrent epileptiform discharges. Therefore, the present recorded high expression of mRNA of both CACNA1S Ca^2+^ (Cav1.1) and NMDA confirmed NMDA receptor-mediated elevations in [Ca21] during the induction of epilepsy [79].

*T. harzianum* produces D-aspartate oxidase, which has powerful kinetic properties and oxidizes many substrates, e.g., D-Aspartate, D-glutamate, and NMDA [80]. Moreover, the reduced level of NMDA after treatment with *T. harzianum* may be due to its metabolite content of D-aminoacylase, which oxidizes the neutral D-amino acids’ derivatives [81].

It is well known that Ca^2+^ channels on intracellular organelles or the plasma membranes are directly sensitive to primary (extracellular) or second stimuli (intracellular), e.g., NAADP, which causes Ca^2+^ exit from organelles or triggers Ca^2+^ influx into the cell [82]. The high level of glutamate excitotoxicity loses the cellular ability to keep the transmembrane ion gradients and leads to an anoxic depolarization. In turn, Ca^2+^ influx initiates a huge release of glutamate. Therefore, the glutamate receptors of the NMDA class cause excitotoxic injury of neurons [83]. *Trichoderma* produces three bioactive metabolites: Harzianic acid (HA), hydrophobin (HYTLO1), and 6- pentyl- pyrone (6PP). Possibly, the intracellular Ca^2+^ channels residing in the endomembrane are represented by NAADP receptors [84]. The imbalance in the cytosolic Ca^2+^ levels causes excitotoxicity and cell death [82]. Recently, one of the bioactive metabolites of *Trichoderma*, hydrophobin HYTLO1, was found to trigger an NAADP-mediated Ca^2+^ signaling pathway [85]. In addition, the high level of extracellular ATP is triggered by glutamate via NMDA receptor (NMDAR)-mediated hippocampal neuronal death [86].

Changes in the serotonergic, dopaminergic, glutamatergic, cholinergic, and GABAergic systems of schizophrenic patients’ brains may play a role in the disease’s pathophysiology [87]. The recorded high levels of acetylcholine, DOPA, epinephrine, norepinephrine, and glutamate proposed its role in the pathogenesis of the current epileptic model. Hyperactivity of the dopamine system in the brain of schizophrenic disorder [88], for example, may be correlated with high levels of the D2 class of dopamine receptors. The recorded high level of DOPA in the epilepticus model was parallel to the research of Alam, A. and M. Starr [89], following pilocarpine-induced seizures, and noted high levels of DOPA in cortical, hippocampal, and striatal areas.

Both plants and mammals release neurotransmitters to control various neurological signaling molecules in plant cellular communications [90]. Moreover, it is well known that high amounts of endogenous melatonin (as a neurotransmitter) are released from *T. harzianum*, *T. asperellum*, *T. koningii*, *T. viride*, and *T. longibrachiatum*. Therefore, the release of indoleamines, e.g., melatonin from *T. harzianum* [91], can reduce kainite-induced excitotoxic brain damage [92] and stimulate antiepileptic activity in in vivo experiments [93]. Later, it was reviewed that melatonin alters dopamine receptor affinity, which is associated with the downregulation of neuronal firing and seizure activity [94]. Moreover, two possible mechanisms could explain the low level or inhibition of DOPA after current treatment with *T. harzianum* metabolites by either (1) its melatonin content, which impacts the reduction in the neuronal firing rate in the substantia nigra [91] or by (2) inhibiting the synaptic DOPA release in the cortex and striatum [95].

## 5. Conclusions

Secondary metabolites of *Trichoderma* (ThCF) have cytotoxic activity against LS-174T (a colorectal cancer cell line) and anxiolytic-like activity through a GABAergic mechanism of action and an increase in GABA as an inhibitory amino acid in the selected brain regions and reduced levels of NMDA and DOPA. The present data suggested that ThCF may inhibit intracellular calcium accumulation by triggering the NAADP-mediated Ca^2+^ signaling pathway. Therefore, the present results suggested further studies on the molecular pathway for the possible role of each metabolite of ThCF, e.g., 6- pentyl- α- pyrone (6-PP), harzianic acid (HA), and hydrophobin, as an alternative drug to mitigate the side effects of AEDs.

## Figures and Tables

**Figure 1 life-13-01815-f001:**
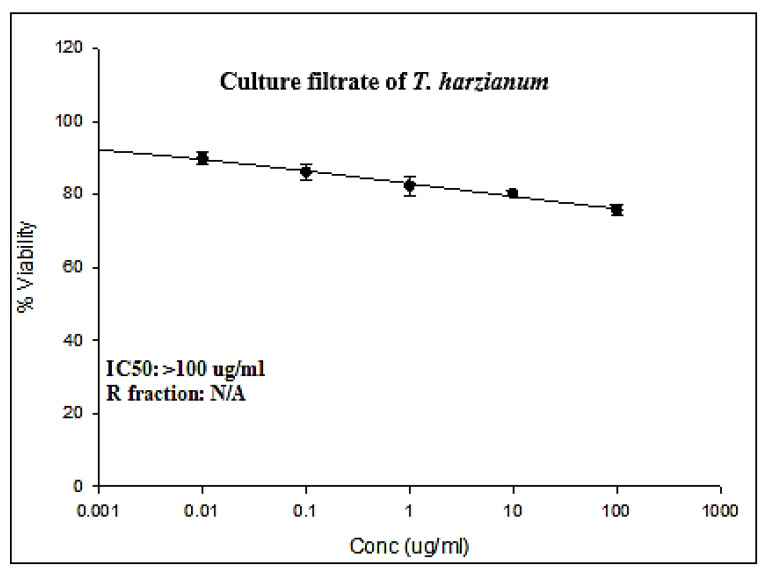
Cytotoxic activities of culture filtrate *Trichoderma harzianum* (ThCF) against LS-174T (colorectal cancer cell line).

**Figure 2 life-13-01815-f002:**
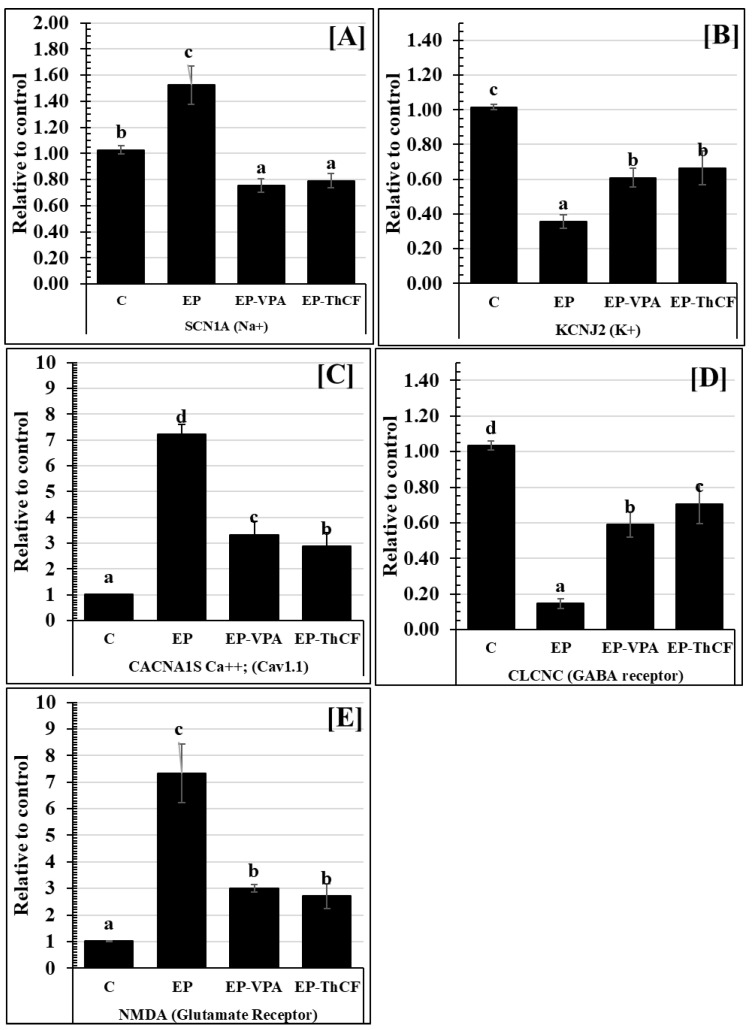
Changes in mRNA gene expression (Relative to control) for some gate receptors Panel (**A**): SCN1A (Na^+^); Panel (**B**): KCNJ2 (K^+^); Panel (**C**): CACNA1S Ca^++^; (Cav1.1); Panel (**D**): CLCNC (GABA receptor), and Panel (**E**): NMDA (Glutamate Receptor) in hippocampal homogenate tissues of different groups: Control (C), epileptic (EP), EP-VPA, and Culture Filtrate of *Trichoderma harzianum*–treated (E-ThCF). Values were represented as Mean ± SD & n = 8 animals. Means within the same parameter and not sharing a common superscript symbol(s) differ significantly at *p* < 0.05.

**Figure 3 life-13-01815-f003:**
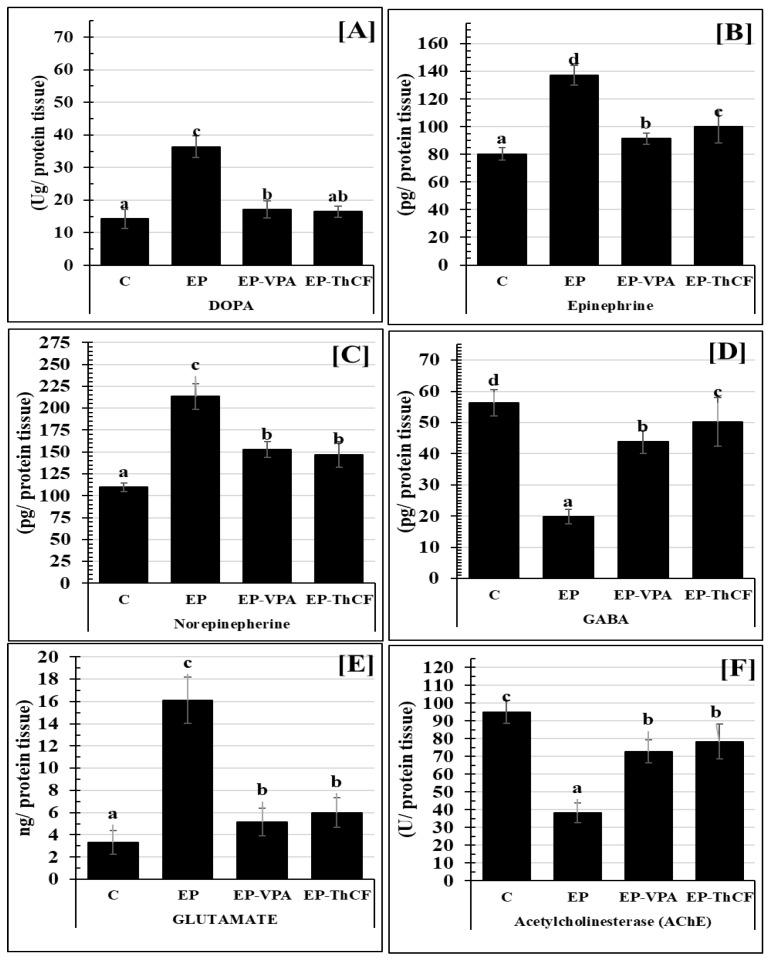
Estimation of some neurotransmitters Panel (**A**): DOPA (Ug/protein tissue); Panel (**B**): Epinephrine (EP) (pg/mg protein tissue); Panel (**C**): Norepinephrine (NE) (pg/mg protein tissue); Panel (**D**): GABA (pg/mg protein tissue); Panel (**E**): Glutamate (ng/mg protein tissue); and Panel (**F**): AChEsterase (U/mg protein tissue), in brain tissues homogenate by ELISA of different groups: Control (C), epileptic (EP), Valproic acid-treated (EP-VPA), and Culture Filtrate of *Trichoderma harzianum*–treated (EP-ThCF). Values were represented as Mean ± SD & n = 8 animals. This means within the same parameter and not sharing a common superscript symbol(s), are differ significantly at *p* < 0.05.

**Table 1 life-13-01815-t001:** Sequence list of specific primers.

Item	Primer	
SCN1A (Na^+^ channel)	F;5-TCATGGCACAGTTCCTGTATC-3	R;5-GCAGTAGGCAATTAGCAGCAA-3
KCNJ2 (K^+^ channel)	F;5-GCAAACTCTGCTTGATGTGG-3	R;5-TCATACAAAGGGCTGTCTTCG-3
CACNA1S (Cav1.1) (Ca^2+^ channel)	F;5-GACATAATTCCCGCTGCCTG-3	R;5-GTTTCCATTCTTCACCCGCC-3
CLCN2 (GABA-receptor)	F;5-CACTGGATAACAACGCCCA-3	R;5-GCAGGGAATGTAGGTCTGG-3
N-methyl-D-aspartate (NMDA) (Glutamate Receptor)	F;5-ACTCCACACTGCCCATGAAC-3	R;5-TTGTTCCCCAAGAGTTTGCTT-3
GAPDH	F;5-GTGAAGGTCGGAGTCAACG-3	R;5-CAATGCCAGCCCCAGCG-3

**Table 2 life-13-01815-t002:** In vitro cytotoxic activity of culture filtrates *Trichoderma harzianum* (ThCF) against colorectal cancer cell line (LS-174T) assessed by SRB assay.

Concentration (µg/mL)	Cell Growth %	% Cell Growth Inhibition	IC50 (µg/mL)
Control (0.00)	100.00 ± 0.00 ^e^	00.00 ± 0.00 ^a^	<100
0.01	89.91 ± 1.03 ^d^	10.09 ± 1.03 ^b^	
0.10	86.22 ± 1.26 ^c^	13.78 ± 1.26 ^c^	
1.00	82.43 ± 1.55 ^b^	17.57 ± 1.55 ^d^	
10.00	80.35 ± 0.43 ^b^	19.64 ± 0.43 ^d^	
100.00	75.81 ± 0.71 ^a^	24.19 ± 0.71 ^e^	

Values are presented as mean ± SEM (n = 3). Significant differences (*p* < 0.05) are indicated by different superscript letters within the same column.

**Table 3 life-13-01815-t003:** Changes in electrolytes in sera (mM/mL) and hippocampal homogenate tissues (mM/mL tissue) of different groups.

	Na^+^		K^+^		Ca^++^		Cl^−^	
	Serum (mM/mL)	Tissues (mM/g. Tissue)	Serum (mM/mL)	Tissues (mM/g. Tissue)	Serum (mM/mL)	Tissues (mM/g. Tissue)	Serum (mM/mL)	Tissues (mM/g. Tissue)
**C**	139.29 ± 0.56 ^c^	22.62 ± 0.62 ^a^	4.63 ± 0.31 ^a^	94.69 ± 7.50 ^c^	1.05 ± 0.18 ^c^	0.30 ± 0.05 ^a^	103.49 ± 2.17 ^a^	123.72 ± 3.37 ^c^
**EP**	130.65 1.37 ^a^	29.47 ± 0.57 ^c^	6.21 ± 0.26 ^b^	66.52 ± 5.85 ^a^	0.37 ± 0.07 ^a^	0.61 ± 0.02 ^c^	110.05 ± 4.43 ^b^	79.86 ± 6.97 ^a^
**EP-VPA**	137.10 ± 1.70 ^b^	20.36 ± 0.92 ^b^	6.01 ± 0.59 ^b^	78.90 ± 3.29 ^b^	0.81 ± 0.05 ^b^	0.29 ± 0.02 ^a^	103.73 ± 1.40 ^a^	99.99 ± 13.16 ^b^
**EP-ThCF**	137.36 ± 1.19 ^b^	22.86 ± 2.59 ^b^	4.65 ± 0.65 ^a^	90.18 ± 5.46 ^c^	0.81 ± 0.14 ^b^	0.34 ± 0.02 ^b^	104.11 ± 2.57 ^a^	105.60 ± 8.25 ^b^
**F value**	80.185	56.689	28.416	45.579	47.408	214.625	11.228	39.461
** *p* ** **<**	0.000	0.000	0.000	0.000	0.000	0.000	0.000	0.000

Control (C), epileptic (EP), *Valproic*-treated (E-VPA), and Culture Filtrate of *Trichoderma harzianum*–treated (EP-ThCF). Values are represented as Mean ± SD & n = 8 animals. Significant differences (*p* < 0.05) are indicated by different superscript letters within the same column.

## Data Availability

The data used in this study are presented in the Results Section.

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
