# Peer review of "Assessment of Biochemical and Neuroactivities of Cultural Filtrate from Trichoderma harzianum in Adjusting Electrolytes and Neurotransmitters in Hippocampus of Epileptic Rats"

_life, 2023, doi:10.3390/life13091815_

Round 1

Reviewer 1 Report (Previous Reviewer 2)

The authors responded my concerns properly.

Reviewer 2 Report (Previous Reviewer 1)

This manuscript is a resubmission of an earlier submission. The following is a list of the peer review reports and author responses from that submission.

Round 1

Reviewer 1 Report

Atef A Abd El-Rahman *, Sally M.A. El-Shafei, Gaber M.G. Shehab,
Lamjed Mansour, Abdelaziz S.A. Abuelsaad, Rania A Gad

Assessment of Biochemical and Neuroactivities of Cultural Filtrate from Trichoderma harzianum in Adjusting Electrolytes and Neurotransmitters in Hippocampus of Epileptic Rats

COMMENTS FOR THE AUTHOR:

1. The presented research is an original and important for
pharmacology and neurobiology. The manuscript is included all parts which needs for thou publication: Abstract, Introduction, Materials and Methods, Results, Discussion, References.

2. The title clearly and precisely reflects the findings of the manuscript.

3. Abstract is it really a summary, include key findings and have an appropriate length.

4. The introduction is written clear, it really introduce and put into perspective of the research.

This study and its introduction, in particular, give an idea of such a socially significant phenomenon as epilepsy. The authors consider the mechanisms of this phenomenon, primarily K-channels, and consider ways to influence the functioning of these channels, on calcium homeostasis. They found an opportunity to influence these systems through the membrane structure using mushroom extracts. On the other hand, they showed the anxiolytic activity of extracts by the gabaergic mechanism. It is proposed to use extracts as an alternative drug to mitigate the side effects of antiepileptic drugs.

5. The authors described the methods briefly, but they are sufficient for reproduction by other researchers. There is no need for additional materials.

6. The general logic of the results is correct, the pictures suggested and located strictly in accordance with the sсheme of section. In my opinion, additional experiments are not required.

I didn't find section 3.2.

7. In discussions in detail examine the problems of contextual memory and its role in the preservation of the species. However, in this aspect, it seems to me that the interaction of the sexes is not sufficiently covered, since this also plays a great role for survival. Maybe it is necessary to add consideration of other authors.

8. The figures correspond to the article’s structure; legends to him explain the drawings. The citation is appropriate, the included the basic publications on the topic.

9. Final comments.

The manuscript is fully consistent to the stated theme. I recommend for publication without additions and corrections.

Author Response

Author's Reply to the Review Report

Manuscript title: “Assessment of Biochemical and Neuroactivities of Cultural Filtrate from Trichoderma harzianum in Adjusting Electrolytes and Neurotransmitters in Hippocampus of Epileptic Rats”

Manuscript no: life-2412870

 Author's Reply to the Review Report (Reviewer 1)

COMMENTS FOR THE AUTHOR:

  • I didn't find section 3.2.
  • Heading no. is corrected

Reviewer 2 Report

This is an interesting study aims to investigate the effect of secondary metabolites of ThCF on the expression of neurotransmitters and electrolytes in epileptic neurons, and to investigate the potential effect of ThCF  to mitigate the side effects of antiepileptic drugs. However if the authors would like to see whether ThCF can mitigate the side effects of VPA, the treatment of VPA and ThCF must be included in the experimental design. Furthermore, it looks like that ThCF has a number of effects on epileptic neurons, but the underlying mechanisms are unclear. Further investigations are needed.

Minor editing of English language required

Author Response

Author's Reply to the Review Report

Manuscript title: “Assessment of Biochemical and Neuroactivities of Cultural Filtrate from Trichoderma harzianum in Adjusting Electrolytes and Neurotransmitters in Hippocampus of Epileptic Rats”

Manuscript no: life-2412870

Author's Reply to the Review Report (Reviewer 2)

Comments and Suggestions for Authors

  • However, if the authors would like to see whether ThCF can mitigate the side effects of VPA, the treatment of VPA and ThCF must be included in the experimental design.
  • Authors depended upon the fact that the Trichoderma harzianum is generally regarded as safe, any substance or treatment, including natural extracts, can potentially have effects on animal behavior.
  • Previously, the acute intraperitoneal toxicity/pathogenicity of Trichoderma, was investigated previously by Rees (1992), who administered intraperitoneally injection of Trichodex®, which contains Trichoderma harzianum Rifai strain T-39. He revealed that in male rats, the LD50 for the Trichoderma harzianum was found to be 644 mg/Kg, while it was 1087 mg/Kg in female rats. The combined male and female rat results were 806 mg/Kg. In addition, there were no signs of substantial deleterious effects at the lowest dose, 1.5 x 107 cfu/animal. This trial was considered acceptable as an alternative to an intravenous study using fungi as active pesticides.
  • Furthermore, it looks like that ThCF has a number of effects on epileptic neurons, but the underlying mechanisms are unclear. Further investigations are needed.
  • Actually, the current study is a one out three parts that investigate the amerolative effects of Trichoderma harizanum, whereas, other parts related study damaged histoarchitecture of the brain in epileptic rats by assessing seizure intensity scale and behavioral impairments and follow up the spontaneous motor seizures during status epilepticus phases in rats.
  • Comments on the Quality of English Language: Minor editing of English language required
  • The manuscript is revised and improved through language editing service by English Department, College of Art, Beni-Suef University for language and grammar, from the aspect of fluency, and share their experience.

Reviewer 3 Report

Title: Assessment of Biochemical and Neuroactivities of Cultural Filtrate from Trichoderma harzianum in Adjusting Electrolytes and Neurotransmitters in Hippocampus of Epileptic Rats

The manuscript titled “Assessment of Biochemical and Neuroactivities of Cultural Filtrate from Trichoderma harzianum in Adjusting Electrolytes and Neurotransmitters in Hippocampus of Epileptic Rats” details a very interesting study indicating the effect of secondary metabolites of Trichoderma harzianum cultural filtrate (ThCF) on the expression of different neurotransmitters and electrolytes in epileptic neurons, and to investigate the potential effect of ThCF to mitigate the side effects of antiepileptic drugs. Results of this research indicated that ThCF is able to ameliorate hyponatremia, hyperkalemia, and hypocalcemia in sera of epilepticus model.  It also improved the elevated levels of many neurotransmitters and inhibited the release of GABA concentration and acetylcholine-esterase activity. Expression of mRNA for some gate receptors are also ameliorated by ThCF hippocampal homogenate tissues. ThCF ameliorates the downregulation of mRNA gene expression for some gate receptors (SCN1A; CACNA1S and NMDA) in hippocampal homogenate tissues.

Abstract:

1) Title of manuscript and aim of study in abstract are different.

2) Context of the study is too long.

3) There is no statically significance in the abstract section (or comparison with the control group).

4) Effective dose/concentration is not mentioned in the conclusion of abstract.

5) Conclusion is very vague

Introduction:

1) write short note on GABA concentration and acetylcholine-esterase activity in relationship with status epilepticus, epileptogenesis and epileptic seizures. 

2) No information is provided about of hyponatremia, hyperkalemia, and hypocalcemia in the pathophysiology of epilepsy.

3) Please add more description about the role of SCN1A, CACNA1S and NMDA receptors in the introduction section.

Materials and methods:

1) How many animals used for studies and? Add the sample size (n per group). Was this number accepted by the Ethical Committee?

2) Were all seizures monitored by research staff, and/or did staff intervene at any points (e.g. based on seizure duration, stage, or animal appearance/behaviour) to alleviate suffering and distress, and/or to assist in terminating the seizure? If not, please explain why, and indicate whether the ethics committee specifically approved this aspect of the study design.

3) “The rats were thought to experience seizure episodes when generalised seizure activity was constantly shown without normal behaviour during each episode. When the rats experienced seizures every 2–5 minutes for a continuous hour, they took place”. Did the Authors observe tonic-clonic seizures or motor seizures? How tonic-clonic seizures and motor seizures were observed? By using video recording? What is the difference between motor seizures and tonic-clonic seizures? Pilocarpine-induced convulsion: were all the animals presented tonic-clonic seizures? If no, the authors should provide the percentage or number of animals in each group that did not reach stage 5 seizure (in Racine’s scale or seizure stage).

Results:

1) You must to determine the dose-response curves (DRC) of various the studied medicinal plant and standard drugs (valproic acid in your experimental conditions), respectively for each anticonvulsant test. If you want to investigate the neuroprotective effects of the ThCF, you have to compare ED50 and efficacy; and to demonstrate which extracts is more effective. The study would be greatly improved by including the DRC.

2) Statistical Rigor: The results section does not reach the current standards for publication. All data should include error bars and statistical evaluation of the data with N and P values indicated in the figure legends. Results should be reported, in the manuscript, according to current statistical nomenclature, including p-values, degrees of freedom, Chi-square values (H), F- and/or t- values. Only p-values are NOT acceptable. You are encouraged to assess the data with statistical rigor and submit your manuscript again. For each swimming model used, include the dependent measure used for the statistical analyses in the results section. The authors should provide the degrees of freedom used in the ANOVA considering the following format F (degrees of freedom: two values) = value, p<value.

Discussion:

1) I am of the view that the discussion section of this article should be treated with more details and consistency. Readers might expect your discussion to be extended to possible explanations or justifications of the activities found. Based on previous knowledge, postulates can be posed in case no scientific arguments are available to explain the results obtained. It is always important to compare your results with previous findings and refer to similar works in the field.

2) Clarify, elaborate and discuss the active constituents of Trichoderma harzianum cultural filtrate (ThCF).

Conclusions:

1) Conclusion is very vague.

2) Check for grammatical mistakes throughout the text according to the instruction for authors.

Level of Interest

An article of importance in its field.

Check for grammatical mistakes throughout the text according to the instruction for authors.

Author Response

Author's Reply to the Review Report

Manuscript title: “Assessment of Biochemical and Neuroactivities of Cultural Filtrate from Trichoderma harzianum in Adjusting Electrolytes and Neurotransmitters in Hippocampus of Epileptic Rats”

Manuscript no: life-2412870

Author's Reply to the Review Report (Reviewer 3)

 Abstract:

  • Title of manuscript and aim of study in abstract are different.
  • Adjusted
  • Context of the study is too long.
  • Adjusted
  • There is no statically significance in the abstract section (or comparison with the control group).
  • Summary section is modified to committed with standard 250 words of the journal qualifications and standard form
  • Effective dose/concentration is not mentioned in the conclusion of abstract.
  • It is now mentioned
  • Conclusion is very vague
  • Conclusion: Secondary metabolites of Trichoderma (ThCF) have anxiolytic-like activity through a GABAergic mechanism of action and an increase of GABA as inhibitory amino acid in the selected brain regions and reduced level of NMDA and DOPA. The present data suggested that ThCF may inhibits the intracellular calcium accumulation by triggering the NAADP-mediated Ca2+ signaling pathway. Therefore, the present results suggested further studies on the molecular pathway for each metabolite of ThCF e.g. 6- pentyl- α- pyrone (6-PP), harzianic acid (HA), and hydrophobin as an alternative drug to mitigate the side effects of AEDs.

Introduction:

  • write short note on GABA concentration and acetylcholine-esterase activity in relationship with status epilepticus, epileptogenesis and epileptic seizures. No information is provided about of hyponatremia, hyperkalemia, and hypocalcemia in the pathophysiology of epilepsy. Please add more description about the role of SCN1A, CACNA1S and NMDA receptors in the introduction section.

Concise literature was added in page 2, lines 61-85

 Materials and methods:

  • How many animals used for studies and? Add the sample size (n per group). Was this number accepted by the Ethical Committee?
  • The current manuscript uses the minimum sample size formula applied in M. Sullivan, Fundamentals of Statistics, 2nd ed., Upper Saddle Creek, NJ: Pearson Education, Inc., 2008 p. 414. Moreover, authors depend upon their previous studies for own lab that were published as follows:

Abd Allah HN, Abdul-Hamid M, Mahmoud AM, Abdel-Reheim ES. Melissa officinalis L. ameliorates oxidative stress and inflammation and upregulates Nrf2/HO-1 signaling in the hippocampus of pilocarpine-induced rats. Environ Sci Pollut Res Int. 2022 Jan;29(2):2214-2226. doi: 10.1007/s11356-021-15825-y. Epub 2021 Aug 7. PMID: 34363578.

Gad, R. A., Abdel-Reheim, E. S., Ebaid, H., Alhazza, I. M., & Abuelsaad, A. S. (2022). Mitigating effects of Passiflora incarnata on oxidative stress and neuroinflammation in case of pilocarpine-Induced status epilepticus model. Journal of King Saud University-Science, 34(3), 101886.‏

  • Moreover, the quantitative data were collected from six rats in all experimental groups.

  • Were all seizures monitored by research staff, and/or did staff intervene at any points (e.g. based on seizure duration, stage, or animal appearance/behaviour) to alleviate suffering and distress, and/or to assist in terminating the seizure? If not, please explain why, and indicate whether the ethics committee specifically approved this aspect of the study design. “The rats were thought to experience seizure episodes when generalized seizure activity was constantly shown without normal behavior during each episode. When the rats experienced seizures every 2–5 minutes for a continuous hour, they took place”. Did the Authors observe tonic-clonic seizures or motor seizures? How tonic-clonic seizures and motor seizures were observed? By using video recording? What is the difference between motor seizures and tonic-clonic seizures? Pilocarpine-induced convulsion: were all the animals presented tonic-clonic seizures? If no, the authors should provide the percentage or number of animals in each group that did not reach stage 5 seizure (in Racine’s scale or seizure stage).
  • Modified in method section as follows: Experimentally induction of epilepsy was done (Abdel-Reheim 2009). Briefly, the experimental rats received intraperitoneal injections of methylscopolamine (1 mg/kg) for 30 minutes before treatment with 300 mg/kg of pilocarpine hydrochloride. After each injection the convulsive behavior was observed for 30 min, and resultant seizures were scored according to (Eraković, Župan et al. 2001) as follows: stage 0, no response; stage 1, ear and facial twitching; stage 2, convulsive waves axially through the body; stage 3, myoclonic jerks and rearing; stage 4, clonic convulsions with the animal falling on its side; stage 5, repeated severe tonic–clonic convulsions or lethal convulsions. The animals were considered to be kindled after receiving pilocarpine hydrochloride injections and having exhibited at least three consecutive stage 4 or 5 seizures (Becker, Grecksch et al. 1995). Rats experiencing lethal convulsions and those who did not develop kindling were excluded from the study. Diazepam (4 mg/kg, i.p.) was given as needed, every 20 minutes, to stop seizures.

Abdel-Reheim, E. S. (2009). "Physiological and biochemical studies on the melatonin effect on the fertility of epileptic rats. ." J. Egyp. Ger. Soci. Zool. 58: 1-25.

Becker, A., G. Grecksch and H. Schröder (1995). "Nω-nitro-l-arginine methyl ester interferes with pentylenetetrazol-induced kindling and has no effect on changes in glutamate binding." Brain research 688(1-2): 230-232.

Eraković, V., G. Župan, J. Varljen, J. Laginja and A. Simonić (2001). "Altered activities of rat brain metabolic enzymes caused by pentylenetetrazol kindling and pentylenetetrazol—induced seizures." Epilepsy Research 43(2): 165-173.

Abd Allah HN, Abdul-Hamid M, Mahmoud AM, Abdel-Reheim ES. Melissa officinalis L. ameliorates oxidative stress and inflammation and upregulates Nrf2/HO-1 signaling in the hippocampus of pilocarpine-induced rats. Environ Sci Pollut Res Int. 2022 Jan;29(2):2214-2226. doi: 10.1007/s11356-021-15825-y. Epub 2021 Aug 7. PMID: 34363578.

Al-Bishri, W. M., Abdel-Reheim, E. S., & Zaki, A. R. (2017). Purslane protects against the reproductive toxicity of carbamazepine treatment in pilocarpine-induced epilepsy model. Asian Pacific Journal of Tropical Biomedicine, 7(4), 339-346.‏

Gad, R. A., Abdel-Reheim, E. S., Ebaid, H., Alhazza, I. M., & Abuelsaad, A. S. (2022). Mitigating effects of Passiflora incarnata on oxidative stress and neuroinflammation in case of pilocarpine-Induced status epilepticus model. Journal of King Saud University-Science, 34(3), 101886.‏

  • During quantifying seizure episodes, authors depend only on their own observation for animal’s behavior and scoring the seizure, which noticed its severity and assess the role of each treatment or intervention in addition to explain many of the studied biochemical and histological studies. All such observations and scoring were done individually, by the aid of 1st, 2nd, 5th and 6th
  • Furthermore, our research lab's current infrastructure lacks many instruments to record the epileptic episode per each animal, as we lack video monitoring, which provides a more accurate and detailed assessment of seizure activity, and electroencephalography (EEG), which measures the electrical activity of the brain and is a valuable tool for quantifying seizures.

 Results:

  • You must to determine the dose-response curves (DRC) of various the studied medicinal plant and standard drugs (valproic acid in your experimental conditions), respectively for each anticonvulsant test. If you want to investigate the neuroprotective effects of the ThCF, you have to compare ED50and efficacy; and to demonstrate which extracts is more effective. The study would be greatly improved by including the DRC.
  • The concentration of Trichoderma harzianum in the suspension of the cultured filtrate refers to the amount or density of harzianum present in the liquid after filtration. The specific concentration can vary depending on the cultivation conditions and the intended application. Typically, it is measured in terms of colony-forming units (CFUs) per milliliter (CFU/ml) or other relevant units of measurement. The ideal concentration of Trichoderma harzianum colony-forming units (CFUs) per milliliter (CFU/ml) was around 10^6-10^8 CFU/ml is often considered suitable for many agricultural, biological and biocontrol purposes. Generally, the concentration of Th-CF in the suspension during the final part of the extraction designated as 100% concentration (Moo-Koh et al., 2022). The current manuscript depends upon previous studies and gel filtration chromatography analysis performed by one of authorship’s Ph.D. thesis (El-Shafei, 2015), whereas, she mentioned that the predominant secondary metabolites in the ThCF have a protein nature (Peptaibols). A recent investigation confirmed that Peptaibols are the primary secondary metabolites in ThCF. (Barakat et al., 2023). So, the protein concentration in the filtrate was measured and found to be 600 mg/ml. Therefore, the dosage for the animals was 0.5 ml ThCF (300 mg protein/kilogram of body weight.
  • References

El-Shafei, S.M.A. (2015). Characteristics of the biological effects of Nigella sativa, Salvia officinalis and Trichoderma on the living systems. Ph.D. thesis (written in Russian) award by the department of biochemistry, institute of fundamental medicine and biology, Kazan (Volga region) federal university, Kazan, Russian federation.

Moo-Koh, F.A.; Cristóbal-Alejo, J.; Andrés, M.F.; Martín, J.; Reyes, F.; Tun-Suárez, J.M.; Gamboa-Angulo, M. In Vitro Assessment of Organic and Residual Fractions of Nematicidal Culture Filtrates from Thirteen Tropical Trichoderma Strains and Metabolic Profiles of Most-Active. J. Fungi 2022, 8, 82

Barakat I, Chtaina N., El Kamli T, Grappin P., El Guilli M., Ezzahiri B. (2023). Bioactivity of Trichoderma harzianum A peptaibols against Zymoseptoria tritici causal agent of septoria leaf blotch of wheat. Journal of Plant Protection Research. 63: 59–67

  • Statistical Rigor: The results section does not reach the current standards for publication. All data should include error bars and statistical evaluation of the data with N and P values indicated in the figure legends. Results should be reported, in the manuscript, according to current statistical nomenclature, including p-values, degrees of freedom, Chi-square values (H), F- and/or t- values. Only p-values are NOT acceptable. You are encouraged to assess the data with statistical rigor and submit your manuscript again. The authors should provide the degrees of freedom used in the ANOVA considering the following format F (degrees of freedom: two values) = value, p<value.

  • Done, already all F values were added in table (2). In addition, means for groups in homogeneous subsets are displayed in result description pages 9 & 10. Degree of freedom was 3.

Discussion:

  • I am of the view that the discussion section of this article should be treated with more details and consistency. Readers might expect your discussion to be extended to possible explanations or justifications of the activities found. Based on previous knowledge, postulates can be posed in case no scientific arguments are available to explain the results obtained. It is always important to compare your results with previous findings and refer to similar works in the field.
  • However, there is no research on Trichoderma's impact on the expression of several neurotransmitters and electrolytes in neurons with epilepsy. Generally, natural products like fungi belonging to the Trichoderma genus are well-known producers of secondary metabolites. Therefore, the current research depends on testing the effectiveness of the general compound of the culture filtrate as the first real experiment in the field of epileptogenesis in animal model, even some other types of fungi have been tried before. It is expected that some real active compounds will be purified and extracted, to be evaluated throughout delineating their potential role on the GABA and acetylcholine-esterase activity in relationship with status epilepticus, epileptogenesis and epileptic seizures, in addition to their role on the hyponatremia, hyperkalemia, and hypocalcemia during the pathophysiology of epilepsy.

  • Clarify, elaborate and discuss the active constituents of Trichoderma harzianumcultural filtrate (ThCF).
  • Actually, some of the effective active constituents according to the previous literature was mentioned in Page 11 (lines 408-414) and Page 12 (lines 441-445; 451-458; 472-279)

 Conclusions:

  • Conclusion is very vague. revised
  • Check for grammatical mistakes throughout the text according to the instruction for authors.
  • Comments on the Quality of English Language: Check for grammatical mistakes throughout the text according to the instruction for authors. Done

Round 2

Reviewer 2 Report

The authors did not respond my concerns properly.

Reviewer 3 Report

Title: Assessment of Biochemical and Neuroactivities of Cultural Filtrate from Trichoderma harzianum in Adjusting Electrolytes and Neurotransmitters in Hippocampus of Epileptic Rats

Reviewer’s report:

Authors have addressed almost all the issue I raised out from the initial manuscript. I think that the article can be published at its present state.

 Major compulsory revisions: None

Minor essential revisions: None

Discretionary Revisions: None

Level of interest: An article of importance in its field

Quality of written English: Acceptable

Declaration of competing interests: I have no competing/conflicting interest whatsoever.
